# Study of a Gate-Engineered Vertical TFET with GaSb/GaAs_0.5_Sb_0.5_ Heterojunction

**DOI:** 10.3390/ma14061426

**Published:** 2021-03-15

**Authors:** Haiwu Xie, Yanning Chen, Hongxia Liu, Dan Guo

**Affiliations:** 1Key Laboratory for Wide-Band Gap Semiconductor Materials and Devices of Education, The School of Microelectronics, Xidian University, Xi’an 710071, China; xiehaiwu.love@163.com (H.X.); danguoElen@163.com (D.G.); 2The School of Physics and Electronic Information Engineering, Qinghai Normal University, Xining 810016, China; 3Beijing Engineering Research Center of High-reliability IC with Power Industrial Grade, Beijing Smart-Chip Microelectronics Technology Co., Ltd., Beijing 100192, China; chenyanning@sgitg.sgcc.com.cn

**Keywords:** band-to-band tunneling (BTBT), GaSb/GaAs_0.5_Sb_0.5_ heterojunction, dual material gate, heterogeneous dielectric, lightly doped source-pocket

## Abstract

It is well known that the vertical tunnel field effect transistor (TFET) is easier to fabricate than the conventional lateral TFETs in technology. Meanwhile, a lightly doped pocket under the source region can improve the subthreshold performance of the vertical TFETs. This paper demonstrates a dual material gate heterogeneous dielectric vertical TFET (DMG-HD-VTFET) with a lightly doped source-pocket. The proposed structure adopts a GaSb/GaAs_0.5_Sb_0.5_ heterojunction at the source and pocket to improve the band-to-band tunneling (BTBT) rate; at the same time, the gate electrode is divided into two parts, namely a tunnel gate (M1) and control gate (M2) with work functions Φ_M1_ and Φ_M2_, where Φ_M1_ > Φ_M2_. In addition, further performance enhancement in the proposed device is realized by a heterogeneous dielectric corresponding to a dual material gate. Simulation results indicate that DMG-HD-VTFET and HD-VTFET possess superior metrics in terms of DC (Direct Current) and RF (Radio Frequency) performance as compared with conventional VTFET. As a result, the ON-state current of 2.92 × 10^−4^ A/μm, transconductance of 6.46 × 10^−4^ S/μm, and average subthreshold swing (SS_ave_) of 18.1 mV/Dec at low drain voltage can be obtained. At the same time, DMG-HD-VTFET could achieve a maximum f_T_ of 459 GHz at 0.72 V gate-to-source voltage (V_gs_) and a maximum gain bandwidth (GBW) of 35 GHz at V_gs_ = 0.6 V, respectively. So, the proposed structure will have a great potential to boost the device performance of traditional vertical TFETs.

## 1. Introduction

According to the Moore’s law, both high switching speed and low-power consumption are necessary for nanoscale devices. So, compactness in devices is an urgent problem to be solved in order to match the requirements of Moore’s law; as a result, a drastic reduction in volume and cost of the devices will occur. However, as the conventional metal oxide semiconductor field effect transistors (MOSFETs) scale down to nano regimes, various performance setbacks will occur due to device miniaturization. In addition, the scaling of MOSFETs to the nanometer regime faces big challenges of short channel effects (SCEs) and high leakage currents; meanwhile, the limitation of 60 mV/Dec subthreshold swing (SS) of MOSFETs can not be broken on account of their working mechanism, at the same time, a drain-induced barrier lowering (DIBL) effect is also arising. In recent years, several types of device structures have been proposed and investigated by numerous researchers to overcome these shortcomings. For about 15 years, many of the studies reported have focused on the tunnel field effect transistor (TFET) [1,2,3,4], which is a promising candidate for future ultra-low power applications. Unlike MOSFETs, TFETs generate current by a gate-controlled band-to-band tunneling (BTBT) mechanism, where carriers inject from source to channel with controlling the band bending in the source/channel interface [5,6]. Compared with conventional thermionic emission in MOSFET, TFET employs the quantum mechanical band-to-band tunneling, which helps TFET break the SS limit of 60 mV/Dec and overcome the short channel effects (SCEs) [7,8,9,10]. In addition, the fabrication of TFETs is compatible with CMOS (Complementary Metal-Oxide-Semiconductor Transistor), so there is no need to develop additional processes. Although the leakage current of conventional TFETs is small under the OFF-state condition, a small ON-state current in case of TFET is not impressive enough. Researchers present various approaches to overcome these issues; consequently, a lot of novel structures have been proposed, strained silicon techniques have been adopted, and III–V material TFETs have been investigated. However, adopting lateral abrupt junction at the tunneling interface is a common characteristic in TFETs reported in recent years, which inevitably introduces complex manufacturing processes [11,12,13,14,15]. In order to comprehensively consider the feasibility of the process, reduction of area footprint, and the whole performance, vertical topology TFET is introduced. In vertical TFET (VTFET), an appropriate selection of work function for metal electrodes can improve the trade-off between analog/RF and linearity parameters, so this device offers significantly improved performance in terms of analog/RF and linearity results. Consequently, these devices would be useful for Internet of Things application and low-power circuits.

In the view of fabrication, the VTFET is easier to manufacture than the conventional TFETs [16,17,18,19,20]; the top–down nanofabrication technology can be used in VTFET, which enables precise control over the physical dimensions of the nanowires as well as the size and configuration [21,22]. Moreover, the number of pads could be scaled up to thousands. Furthermore, there are no trapping-related issues and less leakage current, and it is not difficult to form wrap-gated architectures in VTFET. In addition, the structure of conventional VTFET stems from conventional TFET [23,24], and pocket engineering is still adopted in VTFET, which can increase the electric field near the source and source-pocket. In other words, the tunneling process occurs at the source/source-pocket interface, the tunneling rate of the carrier between the source and the source-pocket can be significantly improved by a thin pocket layer under the source region. In addition, heterojunction VTFETs have better performance than homojunction VTFETs [25,26]. According to the principle of band-to-band tunneling, small m*, small Eg, and appropriate ΔΦ are required in the source region (where m* is the effective carrier mass, Eg is the bandgap, and ΔΦ is the energy range over which tunneling can take place), and III-V material heterojunction is selected to form a heterostructure at the source/source-pocket interface in this simulation due to their suitable material performance and high electron mobility [27,28,29,30,31,32,33,34].

In this paper, a novel structure of dual material gate heterogeneous dielectric vertical TFET (DMG-HD-VTFET) with a lightly doped source-pocket is designed and investigated. Compared with our previous work, the technological realization of this structure is greatly improved due to the use of top–down nanofabrication technology, while its performance is similar to other structures [35,36,37]. DMG-HD-VTFET adopts a GaSb/GaAs_0.5_Sb_0.5_ heterostructure at the source/source-pocket interface to improve the band-to-band tunneling (BTBT) current, and it uses intrinsic GaSb as channel material and GaAs as drain material. Still, a heterogeneous gate dielectric formed by high-k dielectric (HfO_2_) and SiO_2_ is employed, where the high-k dielectric corresponds to the source-pocket and SiO_2_ corresponds to the channel, respectively. In addition, the gate electrode is divided into two parts, namely the tunnel gate (M1) and the control gate (M2) with work functions Φ_M1_ and Φ_M2_, where Φ_M1_ > Φ_M2_. Appropriate work functions for the gate electrodes can further improve the ON-state current and suppress the OFF-state current. As a result, DMG-HD-VTFET has superior metrics in DC and RF performance as compared with conventional VTFET and heterogeneous dielectric vertical TFET (HD-VTFET). Simulation results indicate that a higher ON-state current (Ion), higher transconductance (g_m_) and output transconductance(g_ds_), and higher cutoff frequency(f_T_) and gain bandwidth (GBW) will be generated in DMG-HD-VTFET. So, it could be of great interest to future ultra-low power and high-frequency integrated circuit applications.

The process flow of the proposed DMG-HD-VTFET is similar to the conventional vertical nanowires [26,27,38], using a bottom–up technique. The key steps are as follows: first, the drain, channel, source-pocket and source are formed on the substrate by epitaxy in sequence. Then, spaces for dielectric and metal are etched on the left and right sides. Afterwards, SiO_2_ is formed by the atom layer deposition (ALD) method on the left and right spaces. Then, an anisotropic method is used to etch unwanted SiO_2_. Afterwards, we can form HfO_2_ on SiO_2_ by the sputtering method. Then, the redundant dielectric is removed by isotropic etching on the left and right sides. Afterwards, a gate metal stack is evaporated and patterned. At last, the drain contact is formed [39].

The paper includes the following parts: Section 2 introduces the device structures used in this paper and describes the initial device parameters and essential simulation models. Section 3 shows the optimization for DMG-HD-VTFET. Section 4 concludes the paper.

## 2. Methods

Figure 1a–c show the device structure of pocket engineered conventional VTFET, pocket engineered heterogeneous dielectric vertical TFET (HD-VTFET), and pocket engineered dual material gate heterogeneous dielectric vertical TFET (DMG-HD-VTFET). As can be seen from Figure 1a, compared with HD-VTFET and DMG-HD-VTFET, the pocket engineered conventional VTFET uses the same materials in the source and source-pocket region; moreover, the all-Si channel and drain region is adopted in this structure. Figure 1b is the diagram of heterogeneous dielectric vertical TFET (HD-VTFET); it is easy to see that the channel region and the drain region of HD-VTFET are respectively formed by GaSb and GaAs, and the gate dielectric is divided into two parts, which consists of the cascade connection of high-k dielectric (blue, HfO_2_) and SiO_2_ (yellow). As shown in Figure 1c, DMG-HD-VTFET adopts the same structure as HD-VTFET; however, the gate electrode is divided into two parts, namely the tunnel gate (M1) and control gate (M2) with work functions Φ_M1_ and Φ_M2_, where Φ_M1_ > Φ_M2_. Simulation results indicate that the ON-state current and the OFF-state current can be improved simultaneously by selecting the appropriate work functions for the tunnel gate and control gate.

The total height of the three devices is 85 nm, and the source length of the three devices is 10 nm. The length of the high-k dielectric (the blue part under the gate electrode) is 5 nm, and the length of SiO_2_ (the yellow part under the gate electrode) is 20 nm in HD-VTFET and DMG-HD-VTFET. The length of the gates in VTFET and HD-VTFET are all 25 nm, while the length of the tunnel gate is 5 nm and the length of the control gate is 20 nm in DMG-HD-VTFET. The work functions of gate (Φ_G_) in VTFET and HD-VTFET are all 4.4 eV, while the work function of the control gate (Φ_M2_) is 4.1 eV, and the work function of the tunnel gate (Φ_M1_) is 4.4 eV in DMG-HD-VTFET. The three devices have the same concentration profile: the source doping concentration is 2 × 10^19^ cm^−3^, the source-pocket doping concentration is 1 × 10^19^ cm^−3^, the channel doping concentration is 5 × 10^16^ cm^−3^, and the drain doping concentration is 5 × 10^18^ cm^−3^. The detailed parameters of conventional VTFET, HD-VTFET, and DMGE-HJLTFET used in the simulation are listed in Table 1.

All the simulations of the three devices are carried out using ATLAS Silvaco TCAD version 5.20.2.R. The Atlas local models can not take into account the spatial variation of the energy bands, so the nonlocal band-to-band tunneling model (BBT.NONLOCAL) is used to consider the spatial variation of the energy bands and model the forward and reverse tunneling currents of degenerately doped p-n junctions in case of TFET. A quantum confinement model given by Hansch (HANSCHQM) is also used to consider the confinement effects due to the high doping levels in the source region and the thinner oxide under the gate metal. Moreover, the Schenk model for trap-assisted tunneling (SCHENK.TUNN) is also involved to include the tunneling of electrons from the valence band to the conduction band through trap or defect states and phonon-assisted tunneling effects. To account for the minority carrier recombination effects using concentration-dependent lifetimes, Shockley–Read–Hall related to concentration (CONSRH) is activated. Fermi Statistics (FERMI) and the band gap narrowing (BGN) model are also activated to include the certain properties of a very highly doped region [29,30,31].

## 3. Results and Discussion

This section analyzes the effects of arsenic composition in GaAs_y_Sb_1−y_ on the transfer characteristics in DMG-HD-VTFET, the physical mechanism of DMG-HD-VTFET, the input characteristics, the output characteristics, the effects of device parameters on the transfer characteristics, and the RF performance.

### 3.1. The Effects of Composition Changes in GaAs_y_Sb_1−y_ on the DMG-HD-VTFET Transfer Characteristics

As discussed in Section 2, conventional VTFET, HD-VTFET, and DMG-HD-VTFET adopt the same device and the same models for the comparability of simulation. In addition, the same source-pocket formed by GaAs_y_Sb_1−y_ is used to improve the ON-state current in the three devices. It is very necessary to study the variation of maximum current value with the different composition of arsenic in GaAs_y_Sb_1−y_, where GaAs_y_Sb_1−y_ (y = 0) equals GaSb and GaAs_y_Sb_1−y_ (y = 1) equals GaAs. Figure 2a shows the effects of composition changes in GaAs_y_Sb_1−y_ on the transfer characteristics, and it is very clear that the maximum current of the proposed device increases with the increase of y until y = 0.5 and deceases with the increase of y after y = 0.5. Figure 2b explains the changes of current from the variation of band energy. As can be seen from this figure, the energy valley value of the conduction band in the source-pocket is minimal under y = 0.5, which indicates that the energy valley value of the conduction band decreases with the increase of y until y = 0.5 and increases with the increase of y after y = 0.5. In fact, tunneling is the electrons injection from the source valance band to the channel conduction band. The smaller energy valley value of the conduction band in the source-pocket represents the larger band-to-band tunneling rate without varying other conditions. So, y = 0.5 in GaAs_y_Sb_1−y_, which makes a larger effective tunneling area compared with the other compositions. As a result, y = 0.5 is chosen as the optimal value in GaAs_y_Sb_1−y_ to ensure a higher ON-state current, and the conclusion is valid for all the three structures.

### 3.2. The Physical Mechanism of DMG-HD-VTFET

Figure 3a–d explain the differences of VTFET, HD-VTFET, and DMG-HD-VTFET in regard to physical mechanisms. Figure 3a shows the variation of the electrons’ nonlocal BTBT tunneling rate; as depicted in this figure, the electrons’ nonlocal BTBT tunneling rate of DMG-HD-VTFET is significantly improved compared with those of VTFET and HD-VTFET, and the tunneling region of DMG-HD-VTFET is very focused around the source/source-pocket interface, which helps DMG-HD-VTFET generate the largest current among the three structures; as a comparison, the tunneling region of VTFET near the source-pocket is relatively wide, and the tunneling rate of HD-VTFET near the heterogeneous dielectric interface is smaller than that of DMG-HD-VTFET. Figure 3b indicates the holes’ nonlocal BTBT tunneling rate, and it is not difficult to find that the holes’ nonlocal BTBT tunneling rate of DMG-HD-VTFET is larger than that of VTFET and HD-VTFET in the whole source region, which is consistent with the tunneling process of electrons. In Figure 3c, the electric field distribution of VTFET, HD-VTFET, and DMG-HD-VTFET is illustrated, and it is noticed that the electric field value of DMG-HD-VTFET is markedly improved by the divided gate at the source/source-pocket interface, and the electric field appears to have a local maximum value near the channel/drain interface due to the drain adopting the GaAs material, which can lower the barrier height at the drain/source interface compared with VTFET. In Figure 3d, the current density of the y-direction (parallel to the gate) in three devices is shown, and it can be found that the current density of DMG-HD-VTFET is significantly improved due to the innovation of the device structure. At the same time, the inset of Figure 3d shows the current density of the x-direction (perpendicular to the gate) in three devices, and it is obvious that the current density of DMG-HD-VTFET in the x-direction is always larger than that of VTFET and HD-VTFET. As a result, the current density of DMG-HD-VTFET is obviously enhanced.

Figure 4a illustrates the ON-state (drain-to-source voltage (Vds) = 0.5 V, gate-to-source voltage (V_gs_) = 1 V) energy band diagram of VTFET, HD-VTFET, and DMG-HD-VTFET, where it is obvious to find that an energy band valley is introduced into both HD-VTFET and DMG-HD-VTFET in a heterogeneous dielectric interface, and the energy band valley of DMG-HD-VTFET is further lowered by the dual material gate. Consequently, the conduction band and valance band of DMG-HD-VTFET at the source/source-pocket interface are very close to each other, and the tunneling distance of DMG-HD-VTFET is much smaller than that of VTFET and HD- VTFET. In other words, DMG-HD-VTFET has a lower tunneling width, which ensures a higher tunneling rate compared with VTFET and HD-VTFET. As a result, the amount of tunneling electrons from the source valance band to the source-pocket conduction band is obviously enhanced in DMG-HD-VTFET. Figure 4b shows the OFF-state (V_ds_ = 0.5 V, V_gs_ = 0 V) energy band diagram of VTFET, HD-VTFET, and DMG-HD-VTFET; as can be seen from this figure, the distance of the source valance band and source-pocket conduction band in DMG-HD-VTFET is large enough, and the band energy valley can produce an extra barrier height in the channel region, resulting in an excellent OFF-state performance in this newly proposed device.

### 3.3. The Input Characteristics

The transfer characteristics (I_d_-V_gs_) of VTFET, HD-VTFET, and DMG-HD-VTFET are compared in Figure 5a. The figure is simulated at Vds (drain-to-source voltage) = 0.5 V, whereas V_gs_ varies from 0 to 1.0 V. It may be noted that as a whole, DMG-HD-VTFET can produce a higher ON-state current than that of VTFET and HD-VTFET. On the one hand, this is due to the use of III-V material in the channel and drain region, which improves the performance of DMG-HD-VTFET compared with VTFET; on the other hand, this is due to the divided gate using dual material metal, which further improves the performance of DMG-HD-VTFET compared with HD-VTFET. Consequently, DMG-HD-VTFET ensures a higher ON-state current than VTFET and HD-VTFET, and the ON-state current at V_gs_ = 1.0 V is 2.92 × 10^−4^ A/μm; however, the corresponding ON-state currents of VTFET and HD-VTFET are 9.25 × 10^−5^ A/μm and 1.88 × 10^−4^ A/μm. To check the switching performance of the devices, the average subthreshold swing (SS_ave_) is given by Equation (1):(1)$$SS_{ave}=\frac{V_{t}-V_{min}}{logI_{Vt}-logI_{min}}.$$

As can be seen from Equation (1), SS_ave_ is extracted from Vmin to Vt. Vmin is the gate voltage where tunneling happens for the first time, and Vt is defined as the threshold voltage where the drain current becomes 1 × 10^−8^ A/μm, Imin is the minimum drain current at Vmin, and I_Vt_ is the drain current at the threshold voltage. The calculation result shows that the SS_ave_ of DMG-HD-VTFET is 18.1 mV/Dec. Moreover, unlike VTFET and HD-VTFET, the proposed structure has almost no bipolar effect on the simulation voltage range.

It is well known that transconductance (g_m_) is a very important parameter to measure the analog performance of the devices, which is defined as the first derivative of the drain current (I_ds_) with respect to V_GS_, and the formula is given by Equation (2) [34]:(2)gm=dIdsdVGS.

Figure 5b shows the g_m_ values of the three devices. It can be seen clearly that DMG-HD-VTFET shows a better gm performance than VTFET and HD-VTFET, and the maximum g_m_ value of DMG-HD-VTFET is 6.46 × 10^−4^ S/μm when V_gs_ = 0.74 V, whereas the maximum g_m_ values of VTFET and HD-VTFET are 2.31 × 10^−4^ S/μm at V_gs_ = 0.82 V and 5.55 × 10^−4^ S/μm at V_gs_ = 0.86 V, respectively.

### 3.4. The Output Characteristics

Figure 6a shows the output characteristics of VTFET, HD-VTFET, and DMG-HD-VTFET, and it is noticed that DMG-HD-VTFET shows better saturation performance compared with VTFET and HD-VTFET. As illustrated in Figure 6a, the maximum drain saturation current of DMG-HD-VTFET under V_gs_ = 0.5 V is 3.47 × 10^−5^ A/μm, whereas the maximum drain saturation currents of VTFET and HD-VTFET in this condition are only 3.70 × 10^−6^ A/μm and 4.15 × 10^−6^ A/μm.

Another important parameter to measure the analog performance of devices is output transconductance (g_ds_), which can be calculated by the first derivative of the drain current (I_ds_) with respect to V_DS_, as shown in Equation (3) [35]:(3)gds=dIdsdVDS.

In Figure 6b, we compare the output transconductance of VTFET, HD-VTFET, and DMG-HD-VTFET at V_gs_ = 1 V. It can be viewed that the g_ds_ of DMG-HD-VTFET is greater than that of VTFET and HD-VTFET. The maximum output conductance of DMG-HD-VTFET is 1.09 × 10^−3^ S/μm at V_ds_ = 0.5 V, which is about 24 times higher than that of VTFET and 17 times higher than that of HD-VTFET, as shown in Table 2.

### 3.5. Effect of Device Parameters on the Transfer Characteristics

Many studies in the literature report [2,32,33] that TFETs with source-pockets have better subthreshold characteristics over TFETs without source-pockets, and the concept of pocket engineering in conventional TFET is still suitable for DMG-HD-VTFET. This section consists of two parts. In the first part, we show the impact of source-pocket thickness on transfer characteristics by comparing the transfer characteristics of DMG-HD-VTFET with different source-pocket thicknesses, the effects of which are discussed in detail, including the ON-state current, OFF-state current, and the SS_ave_. Then, the impact of source-pocket concentration on transfer characteristics is analyzed in the second part. Several representative impurity concentrations are selected to simulate the influence, and the highest source-pocket doping concentration is 2 × 10^19^ cm^−3^, which is equal to the source doping concentration in our simulation.

Figure 7a shows the transfer characteristics of DMG-HD-VTFET with different source-pocket thicknesses. As can be seen from this figure, the ON-state current of DMG-HD-VTFET increases with the increase of source-pocket thickness when H_P_ < 6 nm, and the maximum ON-state current at V_gs_ = 1.0 V increases with the increase of source-pocket thickness under these conditions. The simulation result indicates that the maximum ON-state current of DMG-HD-VTFET increases from 7.47 × 10^−5^ to 2.92 × 10^−4^ A/μm when the source-pocket thickness varies from 1 to 5 nm. Moreover, the subthreshold characteristics are always excellent when H_P_ < 6 nm. However, both the ON-state current and SS_ave_ get bad when H_P_ > 6 nm. The maximum ON-state current of DMG-HD-VTFET decreases by two orders of magnitude when H_P_ = 6 nm and H_P_ = 7 nm. Figure 7b explains the variation of Figure 7a from the electric field distribution with different source-pocket thicknesses. As can be seen from this figure, the maximum electric field at the source/source-pocket interface increases with the increase of the source-pocket thickness when H_P_ < 6 nm. In addition, the electric field of the drain/channel interface decreases with the increase of the source-pocket thickness when H_P_ < 6 nm. However, the maximum electric field at the source/source-pocket is only 1.4 × 10^6^ V/cm when H_P_ = 6 nm; at the same time, one electric field peak occurs at the channel and another electric field peak occurs at drain/channel interface when H_P_ > 6 nm. The reason for the second electric field peak near x = 30 nm is the formation of the depletion region at the source-pocket/channel interface when Hp ≥ 6 nm. In other words, the depletion region can not form at the source-pocket/channel interface when Hp < 6 nm; consequently, there is no electric field peak near x = 30 nm. Meanwhile, the depletion region starts to form at the source-pocket/channel interface when Hp = 6 nm; i.e., an electric field peak appears near x = 30 nm, and the distribution of the electric field in the depletion region follows the electric field variation rule in the PN junction, which severely impedes the generation of the current and the control for the channel. Moreover, the electric field peaks between Hp = 6 nm and Hp = 7 nm are very close; however, the depletion region width at the drain/channel interface is larger than that of the source-pocket/channel interface. Consequently, the variation of current is mainly decided by the drain/channel depletion region; as can be seen from Figure 7b, the maximum electric field of Hp = 7 nm is smaller than that of Hp = 6 nm. So, the ON-state current of Hp = 7 nm is larger than that of Hp = 6 nm. As a consequence, the current change shown in Figure 7a appears.

In order to improve the performance of the proposed device, a structure with a lightly doped pocket under the source region is introduced in DMG-HD-VTFET. Figure 8a shows the impact of the source-pocket concentration on the transfer characteristics; it is very clear that the maximum value of the ON-state current occurs at the concentration of 1.0 × 10^19^ cm^−3^, and the corresponding ON-state current is 2.92 × 10^−4^ A/μm. Figure 8b illustrates the variation of ON-state current and average subthreshold swing (SS_ave_) with different source-pocket concentrations; as can be found in this figure, the optimal source-pocket concentration is 1.0 × 10^19^ cm^−3^ in terms of the ON-state current and SS_ave_.

As mentioned in the Introduction, the gate electrode of DMG-HD-VTFET is divided into two parts, namely the tunnel gate (M1) and the control gate (M2) with work functions Φ_M1_ and Φ_M2_; correspondingly, HfO_2_ and SiO_2_ are adopted under M1 and M2. Next, we discuss the effects of M1 and M2 on transfer characteristics.

Figure 9a–d indicate the impact of the tunnel gate work function (Φ_M1_) on the transfer characteristics, electric field distribution, and band diagram with keeping Φ_M2_ = 4.1 eV. First, the selection of Φ_M1_ is very critical for the OFF-state current, which decreases with the increase of Φ_M1_ and is not acceptable when Φ_M1_ < 4.3 eV, as depicted in Figure 9a. In addition, the ON-state current of DMG-HD-VTFET increases with the increase of Φ_M1_ until Φ_M1_ = 4.4 eV, which shows the fact that the ON-state current will not increase when Φ_M1_ > 4.4 eV. In Figure 9b, the distribution of electric field under M1 and M2 explains the phenomenon in Figure 9a. It is very easy to see in Figure 9b that the value of electric field under M1 and M2 increases with the increase of Φ_M1_. Consequently, the increasing electric field under M1 helps to improve the tunneling probability in this region, while increasing the electric field under M2 will raise the barrier height in the drain/channel interface. So, the maximum current appears at Φ_M1_ = 4.4 eV under the mutual influence of the two factors. Figure 9c,d depict the energy band diagram in the ON-state and OFF-state condition, which further explain the variation of ON-state and OFF-state current in Figure 9a. It is observed that the tunneling distance decreases with the increase of Φ_M1_, and the height of the barrier in the drain/source interface increases with the increase of Φ_M1_, which is consistent with the current change in Figure 9a. In Figure 9d, it can be found that the conduction band valley of the source-pocket aligns with the top of the valence band of the source region when Φ_M1_ < 4.3 eV, which profoundly reveals the unacceptable OFF-state current in Figure 9a when Φ_M1_ < 4.3 eV. Hence, considering all the factors, the optimal value of Φ_M1_ is chosen as 4.4 eV.

Figure 10a–d illustrate the impact of control gate work function (Φ_M2_) on the transfer characteristics, electric field, potential, and OFF-state band diagram with keeping Φ_M1_ = 4.4 eV. Figure 10a shows the transfer characteristics of DMG-HD-VTFET under different values of Φ_M2_, where Φ_M2_ varies from 3.8 to 4.7 eV in a step of 0.1 eV. In Figure 10a, we can find that the maximum ON-state current at V_gs_ = 1.0 V and the OFF-state current at V_gs_ = 0 V increase with the decrease of Φ_M2_; therefore, an appropriate value of Φ_M2_ should be chosen. In order to reveal the dependence of transfer characteristics on the electric field and potential under different Φ_M2_, Figure 10b,c show the distribution of the electric field and potential at the cutline under different Φ_M2_. It can be found through the observation of Figure 10b that the electric field under M1 decreases with the increase of the Φ_M2_, which means that a smaller Φ_M2_ should be selected to obtain larger a tunneling probability in this region; on the contrary, the electric field under M2 increases with the increase of the Φ_M2_, which means that a larger Φ_M2_ should be selected to produce a lower potential in this region. In fact, DMG-HD-VTFET exhibits a higher potential under smaller Φ_M2_ at the channel region and the drain/channel interface, as depicted in Figure 10c, which further explains the variation of transfer characteristics in Figure 10a. Moreover, the inserted diagram in Figure 10b shows that the electric field near the drain region decreases with the increase of Φ_M2_ until Φ_M2_ = 4.1 eV, and the electric field in this region increases with the increase of Φ_M2_ after Φ_M2_ = 4.1 eV, which means a serious drop in ON-state current when Φ_M2_ > 4.1 eV. The magnitude of the OFF-state current at V_gs_ = 0 V in Figure 10a is very high when Φ_M2_ <4.1 eV; Figure 10d reveals this phenomenon from the perspective of the energy band. As shown in Figure 10d, the conduction band of the source-pocket partially aligns with the valance band of source when Φ_M2_ < 4.1 eV; as a result, the OFF-state current is very large due to the tunneling process. So, all things considered, the optimal value of Φ_M2_ is selected as 4.1 eV.

To summarize, the optimal values for Φ_M1_ and Φ_M2_ are selected as 4.4 eV and 4.1 eV in this article, and simulation results show that the dual material gate can significantly influence the performance of the proposed structure. The ON-state current and the OFF-state current can be maintained at a good compromise by choosing appropriate work functions for gate electrodes. So, the work function of the gate electrode can be flexibly selected according to actual demand.

### 3.6. Comparison in Terms of Analog/RF Performance

The study of parasitic capacitance is very important to evaluate the frequency characteristics of the integrated circuits (ICs). As many studies in the literature have shown, the characteristics of C_gg_ (gate capacitance), C_gs_ (gate to source capacitance), and C_gd_ (gate to drain capacitance) are of great significance to estimate the frequency characteristics and analog application ability of devices. Figure 11a,b show the C_gs_, C_gd_, and C_gg_ comparison diagram of the three devices at the frequency of 1.0 × 10^6^ Hz. As can be observed from Figure 11a, the trend of C_gs_ versus V_gs_ in three devices is similar. Moreover, in Figure 11a,b, it is very clear that both C_gg_ and C_gd_ of VTFET and DMG-HD-VTFET remain at a small value when V_gs_ < V_ds_ and increase rapidly with the increasing V_gs_ when V_gs_ > V_ds_; however, the C_gg_ and C_gd_ of HD-VTFET maintain a very small value until V_gs_ = 0.7 V, which will significantly affect the gain bandwidth (GBW) of this device.

The cutoff frequency (f_T_) and gain bandwidth (GBW) are important indicators for evaluating the RF performance of devices. Based on Figure 11a,b, the f_T_ and the GBW of three devices are discussed in following part.

The formula of f_T_ is given by Equation (4), which can be expressed as a ratio of gm to C_gg_ [36]:(4)$$f_{T}=\frac{g_{m}}{2\pi C_{gs}\sqrt{1+2\frac{C_{gd}}{C_{gs}}}}\approx \frac{g_{m}}{2\pi (C_{gs}+C_{gd})}=\frac{g_{m}}{2\pi C_{gg}}.$$

The GBW can be calculated by the ratio of gm to C_gd_ for the DC gain value equal to 10, as shown in Equation (5) [37]:(5)$$GBW=\frac{g_{m}}{2\pi 10C_{gd}}.$$

Figure 12a shows the f_T_ curves of VTFET, HD-VTFET, and DMG-HD-VTFET. Benefitting from structural innovation, the f_T_ values of HD-VTFET and DMG-HD-VTFET are much higher than that of VTFET, and the maximum f_T_ values of HD-VTFET and DMG-HD-VTFET are 452 GHz (V_gs_ = 0.86 V) and 459 GHz (V_gs_ = 0.72 V), respectively; whereas, the maximum f_T_ of VTFET is only 270 GHz (V_gs_ = 0.76). In addition, as can be noticed in Figure 12b, the GBW of HD-VTFET is the largest among the three devices due to the capacitance changes of C_gg_ and C_gd_ described in Figure 11a,b. The maximum GBW values of HD-VTFET and DMG-HD-VTFET are 45.1 GHz (V_gs_ = 0.76 V) and 35 GHz (V_gs_ = 0.60 V), respectively; however, the GBW of VTFET is only 24.3 GHz at V_gs_ = 0.64 V.

## 4. Conclusions

In this paper, we demonstrate a dual material gate heterogeneous dielectric vertical TFET (DMG-HD-VTFET) with a lightly doped source-pocket. For the reference of simulation comparison, the same device size is adopted in VTFET, HD-VTFET, and DMG-HD-VTFET. Meanwhile, a lightly doped pocket under the source region is introduced to improve the subthreshold performance of the vertical TFETs; also, a GaSb/GaAs_0.5_Sb_0.5_ heterojunction is adopted to improve the band-to-band tunneling (BTBT) rate. In addition, a heterogeneous dielectric is used to further enhance the performance of HD-VTFET and DMG-HD-VTFET. In this regard, the gate electrode of DMG-HD-VTFET is divided into two parts to improve the ON-state current and suppress the OFF-state current simultaneously. In our simulation, we systematically research (1) the effects of composition changes in GaAs_y_Sb_1−y_ on the transfer characteristics, the physical mechanism, and the input and output characteristics; (2) the effects of source-pocket thickness and concentration, tunnel gate work function, and control gate work function on transfer characteristics; and (3) the analog/RF performances of the devices. Simulation results indicate that DMG-HD-VTFET and HD-VTFET possess superior metrics in terms of DC and RF performance as compared with conventional VTFET.

## Figures and Tables

**Figure 1 materials-14-01426-f001:**
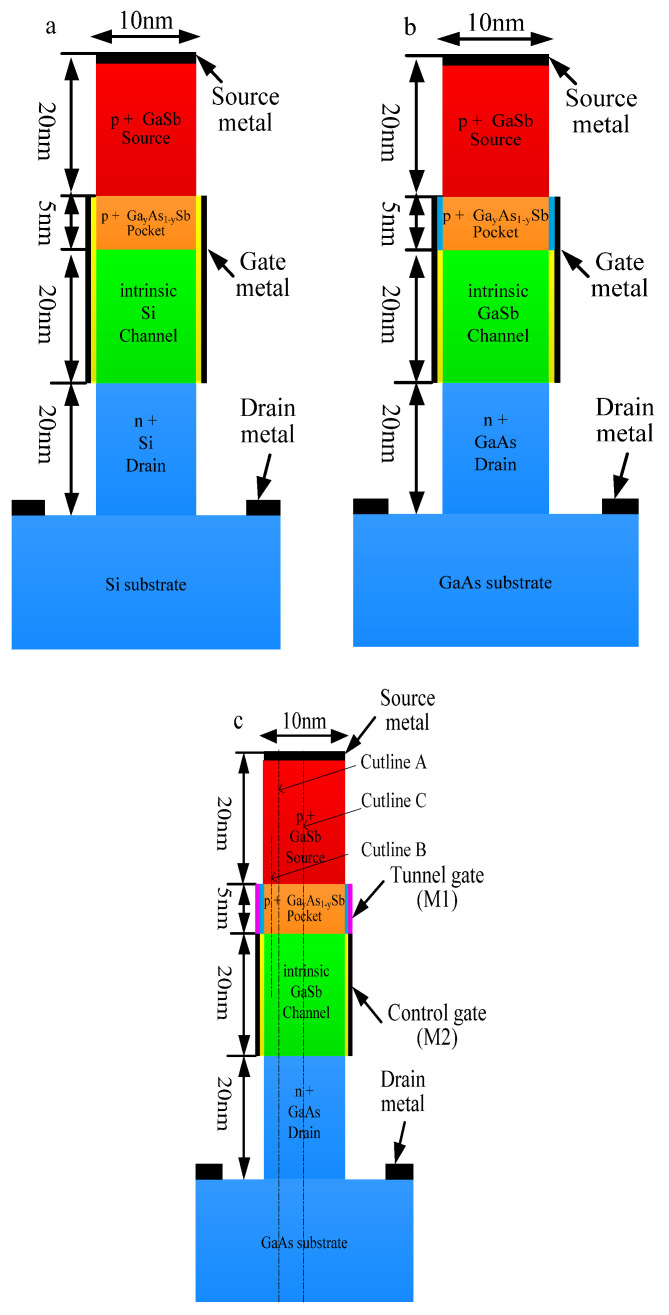
Schematic diagram of (**a**) conventional vertical tunnel field effect transistor (VTFET); (**b**) heterogeneous dielectric vertical TFET (HD-VTFET); and (**c**) dual material gate heterogeneous dielectric vertical TFET (DMG-HD-VTFET).

**Figure 2 materials-14-01426-f002:**
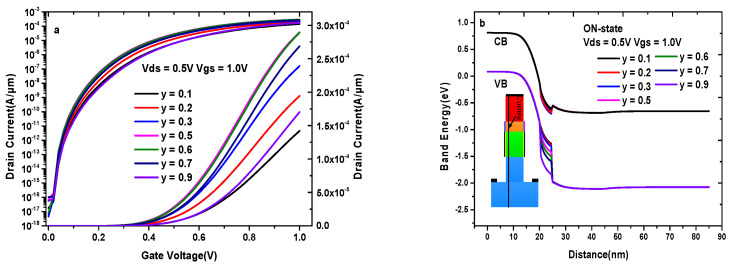
(**a**) The effects of composition changes in GaAsySb1-y on DMG-HD-VTFET transfer characteristics; (**b**) the ON-state energy band diagram of DMG-HD-VTFET with different composition of arsenic in GaAsySb1-y.

**Figure 3 materials-14-01426-f003:**
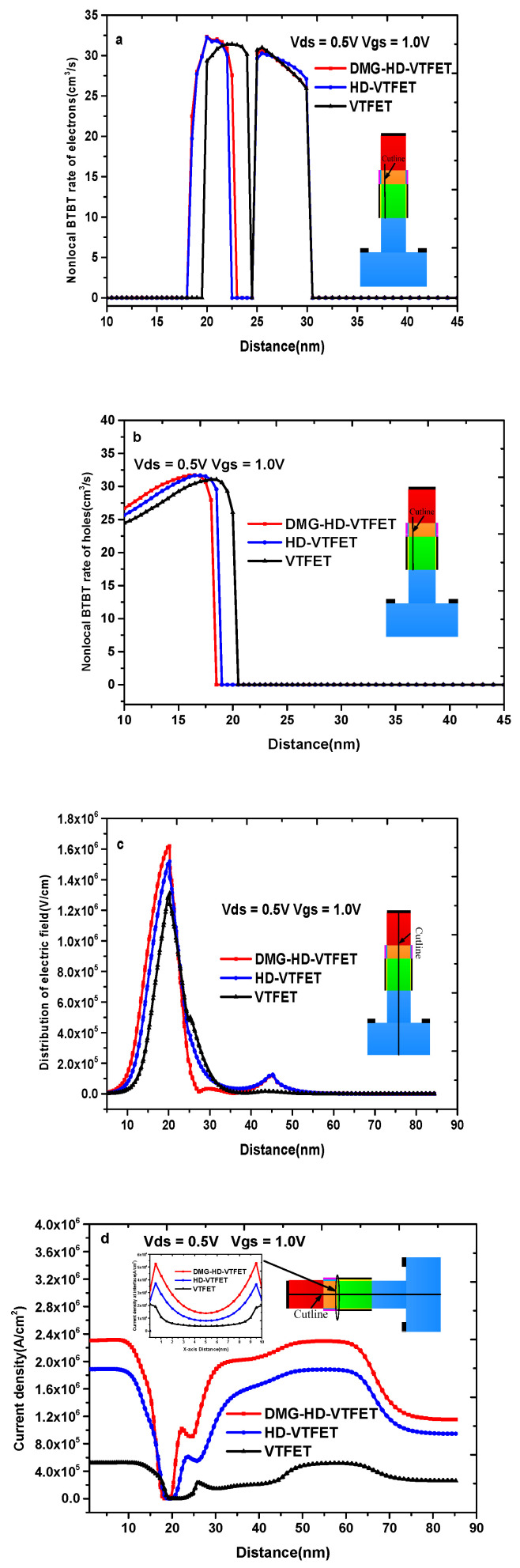
(**a**) Electrons’ nonlocal band-to-band tunneling (BTBT) rate; (**b**) holes’ nonlocal band-to-band tunneling (BTBT) rate; (**c**) electric field distribution; and (**d**) total current density of VTFET, HD-VTFET, and DMG-HD-VTFET.

**Figure 4 materials-14-01426-f004:**
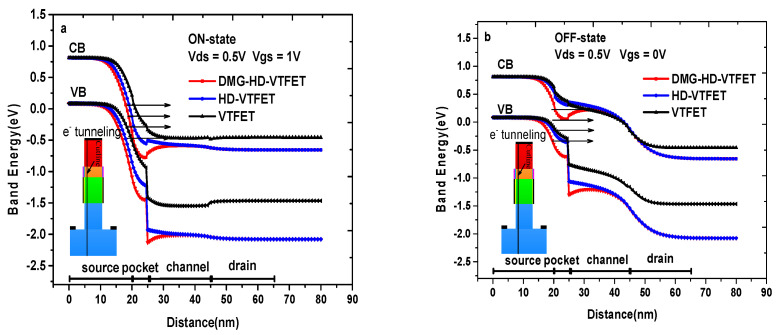
(**a**) The ON-state energy band diagram of VTFET, HD-VTFET, and DMG-HD-VTFET; (**b**) the OFF-state energy band diagram of VTFET, HD-VTFET, and DMG-HD-VTFET.

**Figure 5 materials-14-01426-f005:**
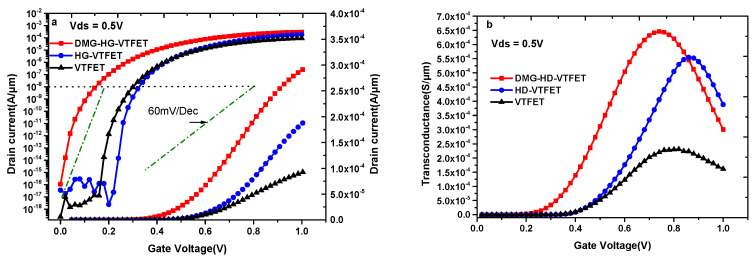
(**a**) The transfer characteristics of VTFET, HD-VTFET, and DMG-HD-VTFET; (**b**) transconductance of VTFET, HD-VTFET, and DMG-HD-VTFET.

**Figure 6 materials-14-01426-f006:**
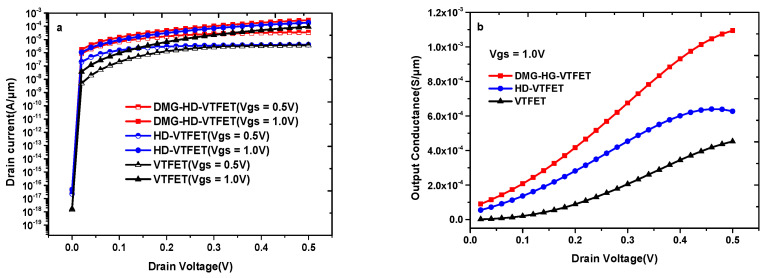
(**a**) Output characteristics of VTFET, HD-VTFET and DMG-HD-VTFET; (**b**) output transconductance of VTFET, HD-VTFET, and DMG-HD-VTFET at V_gs_ = 1 V.

**Figure 7 materials-14-01426-f007:**
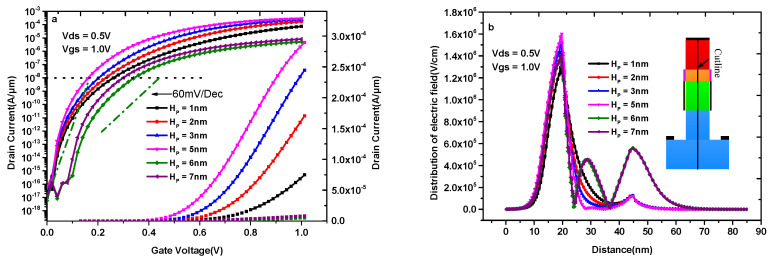
(**a**)Transfer characteristics of DMG-HD-VTFET with different source-pocket thicknesses; (**b**) electric field distribution with different source-pocket thicknesses.

**Figure 8 materials-14-01426-f008:**
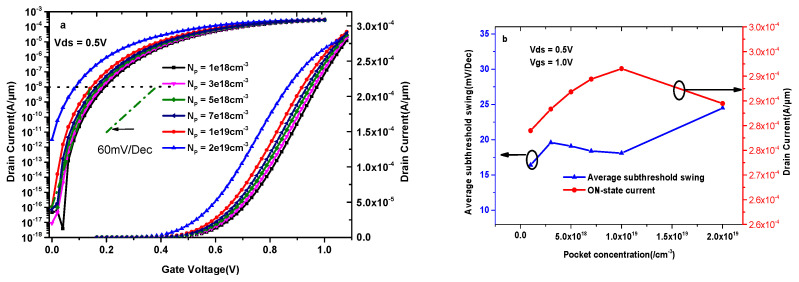
(**a**) Transfer characteristics of DMG-HD-VTFET with different source-pocket concentration; (**b**) the variation of ON-state current and average subthreshold swing (SS_ave_) with different source-pocket concentrations.

**Figure 9 materials-14-01426-f009:**
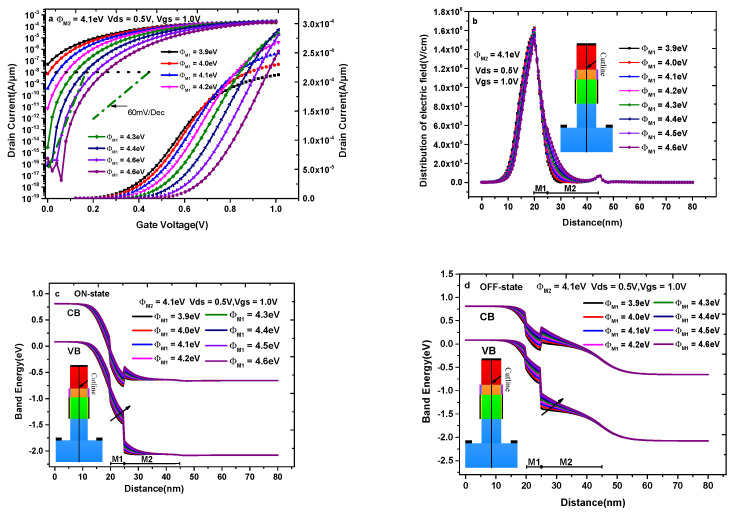
(**a**) Transfer characteristics of DMG-HD-VTFET with different Φ_M1_; (**b**) the distribution of electric field with different Φ_M1_; (**c**) the ON-state energy band diagram with different Φ_M1_; (**d**) the OFF-state energy band diagram with different Φ_M1_.

**Figure 10 materials-14-01426-f010:**
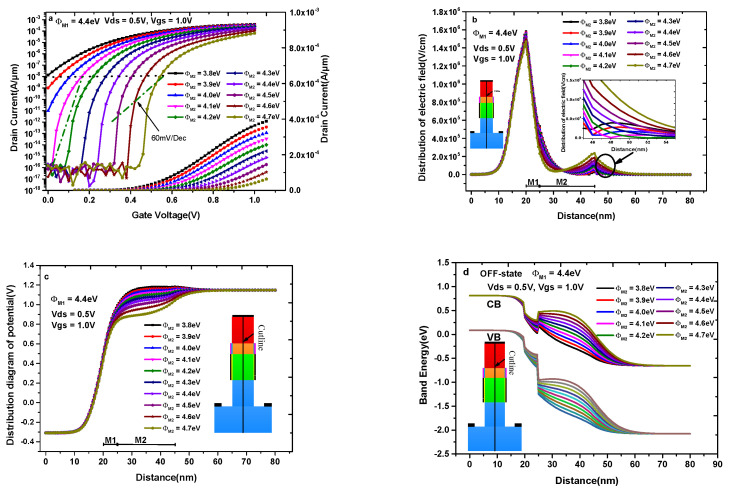
(**a**) Transfer characteristics of DMG-HD-VTFET with different Φ_M2_; (**b**) the distribution of electric field with different Φ_M2_; (**c**) the distribution of potential with different Φ_M2_; (**d**) the OFF-state energy band diagram with different Φ_M2_.

**Figure 11 materials-14-01426-f011:**
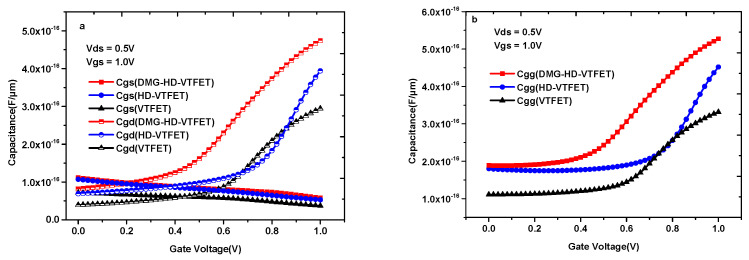
(**a**) Cgs and Cgd comparison diagram; and (**b**) Cgg comparison diagram of VTFET, HD-VTFET, and DMG-HD-VTFET.

**Figure 12 materials-14-01426-f012:**
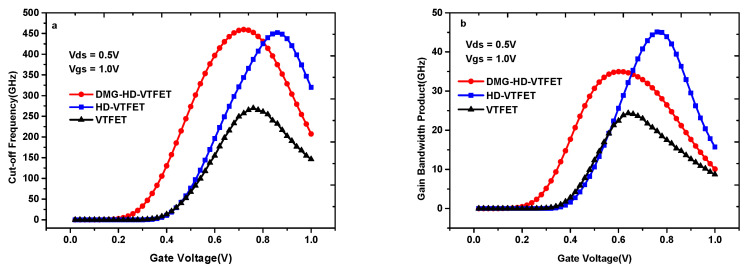
(**a**) The f_T_ curves; and (**b**) the gain bandwidth (GBW) curves of VTFET, HD-VTFET, and DMG-HD-VTFET

**Table 1 materials-14-01426-t001:** The fundamental geometrical and physical parameters of the different devices [19,40].

Parameter Name	VTFET	HD-VTFET	DMG-HD-VTFET
Length of source (L_S_/nm)	10	10	10
Length of substrate (L_SUB_/nm)	30	30	30
Height of source (H_S_/nm)	20	20	20
Height of pocket (H_P_/nm)	5	5	5
Height of channel (H_C_/nm)	20	20	20
Height of drain (L_D_/nm)	20	20	20
Length of gate (H_G_/nm)	25	25	
Length of tunnel gate (H_TG_/nm)			5
Length of control gate (H_CG_/nm)			20
Thickness of SiO_2_ and HfO_2_ (T_OX_/nm)	2	2	2
Length of high-k dielectric (L_HK_/nm)		5	5
Length of SiO_2_ (L_LK_/nm)	25	20	20
Work function of gate (Φ_G_/eV)	4.4	4.4	
Work function of tunnel gate (Φ_M1_/eV)			4.4
Work function of control gate (Φ_M2_/eV)			4.1
Source doping concentration (N_S_/cm^−3^)	2 × 10^19^	2 × 10^19^	2 × 10^19^
Pocket doping concentration (N_P_/cm^−3^)	1 × 10^19^	1 × 10^19^	1 × 10^19^
Channel doping concentration (N_C_/cm^−3^)	5 × 10^16^	5 × 10^16^	5 × 10^16^
Drain doping concentration (N_D_/cm^−3^)	5 × 10^18^	5 × 10^18^	5 × 10^18^

**Table 2 materials-14-01426-t002:** DC characteristics comparison table of VTFET, HD-VTFET, and DMG-HD-VTFET at drain-to-source voltage (V_ds_) = 0.5 V and gate-to-source voltage (V_gs_) = 1.0 V.

	VTFET	HD-VTFET	DMG-HD-VTFET
V_min_ (V)	0.16	0.20	0
V_t_ (V)	0.30	0.33	0.14
Ion (A/μm)	9.25 × 10^−5^	1.88 × 10^−4^	2.92 × 10^−4^
Ioff	1.30 × 10^−17^	2.43 × 10^−17^	1.1 × 10^−16^
Ion/Ioff	7.12 × 10^12^	7.74 × 10^12^	2.66 × 10^12^
SS_ave_ (mV/Dec)	15.8	13.3	18.1
g_m_ (S/μm)	2.31 × 10^−4^	5.55 × 10^−4^	6.46 × 10^−4^
g_ds_ (S/μm)	4.52 × 10^−4^	6.40 × 10^−4^	1.09 × 10^−3^

## Data Availability

The data presented in this study are available on request from the corresponding author.

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
