# Peer review of "Study of a Gate-Engineered Vertical TFET with GaSb/GaAs0.5Sb0.5 Heterojunction"

_materials, 2021, doi:10.3390/ma14061426_

Round 1

Reviewer 1 Report

This work presents an improved heterogeneous dielectric vertical TFET structure, as a modification of the more classic TFET. The modification mainly consists of employing a dual material gate for the source pocket and the channel, respectively. The results are provided by showing comparative TCAD simulation of the new structure with more classical vertical TFET and heterogeneous dielectric vertical TFET. Moreover, the process parameters are swept in detail to assess the impact on the electrical figures of merit. Please consider the following comments:  

In Fig. 1: does 'substract' stand for substrate? Please explicitly indicate M1 and M2 in Fig. 1c.

In Sec. 2, it would be useful to provide additional references on the simulation models adopted. Indeed, they are part of the software, but the reader might benefit from an explicit link to literature. Please make explicit BTBT in the text: band to band tunneling. 

In Sec. 3.1, Fig. 2, for which of the three device structures are these results related to? Are the conclusion valid for all the three structures, as they all share the same GaAs-Sb pocket? Also, it is not clear what is the takeaway for this analysis. Which one is the As composition eventually chosen and why? Please clarify and provide further explanation.

Note on the style: phrases like 'that is to say' or 'what's more' or 'where it is easy to find' or 'As we know' are too colloquial and should be modified. More in general, an revision of English language and style is needed.

In Fig. 6b, could you adopt the same color code as in the other Figures?

It seems to me that not all the symbols in Table 2 are defined within the text.

At the beginning of Sec. 3.5, when the authors mention literature works, please include references for the TFET source-pocket enhancement.

Reviewer 2 Report

Dear,
First, I would like to congratulate the authors for this good piece of work. The proposal is really interesting theoretical and simulation work. But there are some issues I would like to clarify.
   - Along the introduction section I think you have to highlight the increasing relevance, in terms of area-footprint reduction and performance, that are currently acquiring relevance the devices based on vertical topologies. In fact, recently some works have shown the feasability to implement different electronic devices by using this configuration. For instance, single-electron transistors (SET) in a vertical  topology (E. Amat et al; "Influence of Quantum Dot Characteristics on the Performance of Hybrid SET-FET Circuits", IEEE TED, vol. 66 (10), pp. 4461-4467, 2019) and neuronal circuits (J. Robinson et al,; "Vertical nanowire electrode arrays as a scalable platform for intracellular interfacing to neuronal circuits", Nature Nanotechnology, vol. 7, pp. 180-184, 2012).
   - Additionally, in the introduction section I miss some explanation about the usability of this typpe of devices. For instance, would these device be useful for Internet of things, low-power circuits...?
  - Continuing with the previous comment, in some small comparison with results based on lateral TFET ones can be interesting, just to see the achieved improvement. It can be your own simulations or some previous literature work. But to show the enhancement on using vertical topology could give more relevance to your new device topology.
 - All this work is based on device simulation. Then, another interesting thing to analyse it can be the feasability to manufacture it, cannot it? I guess there will be relevant threats into the manufacturing scheme, mainly, the use of the two different gate both metal configurations and gate oxide materials. It would be useful for the reader if you can explain, a little bit, how do you plan to solve these difficulties.
  - For a clear comparison between all the figures where you perform a cross-section along the different VTFETs I think it would be useful to use always the same x-axis, to easily locate the device region; if not can be confusing.
   -Following with the previous comment, when you compare the different device structures, I suggest you to use always the same color for the same device type. Do not change the color at each figure.
   - About the dielectric material you plan two strategies: one with only SiO2 and another with two regions based on high-k and SiO2. Then, have you considered the only use of high-k dielectric for all the gate oxide into your device proposals; because you mix HfO2 with SiO2, why not a full high-k device? Or even you can compare what happen if you exchange the dielectric materials for proposal b and c. Which behavioral difference would you observe?
   - In Fig. 7, the colors of the first and last lines are very similar and ir can introduce a confusing message. Moreover, it is quite confusing to see the behavioral difference between Hp=6 and 7 nm.
   - Additionally, I miss some more physical explanation about the behavior observed when the Hp is larger than 5 nm. Why there appear the two peaks, and which explanation do you have?
   - I miss some reference in section 3.5, as you mention "many literatures report" I need someone.
   - In Table 2, you mention Ion as reference parameter, but it may be more interesting is the Ion/Ioff ratio. 
Finally, I hope you can improve your contribution to achieve a better understanding and evaluation of your work.

 Thanks!

Reviewer 3 Report

- The band gaps of GaAs and GaSb are 1.424 eV and 0.726 eV, respectively, but this difference, in particular for the channel (GaSb) and drain (GaAs) regions, we do not observe in many of the figures where band diagrams are presented (Figures 2b, 4a,b, 9c,d, 10d, ). This should substantially change the entire band diagram. The authors should be more careful and check all the band diagrams, for example, the bandgap for the channel with y=0.1 in Figure 2b is already narrower than the one for GaSb of the source.

- Some misunderstandings between the work functions of the tunnel and control gates, which are characteristic of the chosen dielectric (Table 1), and the variation of the work functions in Figures 9 and 10. Does it mean, the authors have in mind to check the characteristics for different dielectrics to be involved?

- The colors of the gate dielectric in Figure 1 should be explained also in the caption.

- The tunnel gate M1 and the control gate M2 should be depicted in Figure 1c, also providing that we have two electrically separated gate contacts.

- For better scaling transparency of the Distance axis in Figures 2, 3, 4, 7, 9, and 10, a vertical scale/bar corresponding to the Distance axis should be positioned along with one of the structures in Figure 1.

- Related to the previous comment, the “source”, “pocket”, “channel”, and “drain” positions in Figures 4a,b should be presented more precisely, better even with arrows for the exact distance.

- M1 and M2 length is 5 and 20 nm, respectively, however, they do not correspond to the arrows positioning in Figures 9b,c,d, 10b,c.

- The Distance axis in Figures 2b, 3a,b, 4a,b does not correspond to the "Cutline" in the insets.

Technical:

- There are some misunderstandings for Cds and Cgd in Figure 11 legends and caption, also the text “…Figure 11. (a-b) show the Cgs, Cgd and Cgg comparison diagram of the three devices at the frequency of 1.0×106 Hz. As can be observed from Figure 11. (a), the trend of Cgs versus Vgs in three devices is similar. Moreover, in Figure 11.(a-b), it is very clear that both Cgg and Cgd of VTFET and DMG-HD-VTFET remain at a small value when VgsVds; however, the Cgg and Cgd of HD-VTFET maintain…” on Page 12.

- The curves in Figure 8b should have directive arrows for the black/white presentation.

Reviewer 4 Report

The Authors present a simulation study of operation of  gate-engineered vertical TFETs with GaSbGaAsySb1-y source/pocket heterojunction, Si or GaSb channel, Si or GaAs drain. Different gate stacks have been considered. In general three variants of VTFETs have been considered: pocket engineered conventional VTFET, pocket engineered heterogeneous dielectric vertical TFET (HD-VTFET), and pocket engineered dual material gate heterogeneous dielectric vertical TFET (DMG-HD-VTFET).

The simulations have been done in a consistent way. An effect of the pocket parameters as well as gate stack structure and parameters on tunneling, DC performance (IOFF, ION, SS), AC performance (gm, gds, gate capacitances) and RF figures of merit (fT, GBW) have been analyzed. The results have been interpreted via distributions of electric field and band diagrams. It has been shown that HD-VTFETs and DMG-HD-VTFETs are superior with respect VTFETs. Optimum values of the pocket thickness and doping, gate metal workfunctions have been estimated. In this sense the work appears to be practically useful for the device designers.

The reference list is up-to-date, although it is also biased/not objective. A number of recognized papers have not been cited. On the other hand a number of works have been referenced, which have secondary meaning. Also a few unnecessary autocitations have been made. Using [ 27-30] as references for (2), (3), (4), (5) is incorrect !

Detailed comments:

I do not agree with the statement "Recently, many of literatures reported have focused on tunnel field effect transistor (TFET) [1-4]"; the TFETs have been considered for around 15 years;

In Fig.1 the type of the pocket area is incorrectly labelled;

The caption of Table 1 is not precise; in the table there are not only geometrical parameters;

on p.5,6 The statement "As a result, the total current density of DMG-HD-VTFET is obviously enhanced" and Fig.3(d) should be commented, since they show a one-dimensional crosssection of the currend density distribution. This distribution is not constant, what means that the current in the pocker region flows mainly close the gates;

Following this, I think that on p.10 the statemet "It is very easy to see in Figure 9. (b) that the value of electric field under M1 and M2 increases with the increase of ΦM1. Consequently, the increasing electric field under M1 helps to improve the tunneling probability in this region..." is not complete. Since in the pocket area the current flows close to the gates, the changes of the M1 workfunction affect mainly the carrier tunneling to the channel close to the surface; 2-D distributions of current density or tunneling rate, or carrier density would clarify this;

I think that in Fig.9 the M1, M2 regions have been incorrectly marked.

The language of the paper is correct, although pretty often the Authors have used very long, compound satatements which make in some cases the satements not fully clear, e.g.: on p.2 "Moreover the structure of conventional VTFET ... under source region", on p.2 "According to the ... [25-26]", on p.5 "Figure 3. (a) ... DMG-HD-VTFET", on p.6 "Figure 4.(a) ... VTFET and HD-VTVET"

Reviewer 5 Report

In the paper, the authors demonstrate a dual material gate heterogeneous dielectric vertical TFET (DMG-HD-VTFET) with lightly doped source-pocket. The structure proposed by authors adopts GaSb/GaAs0.5Sb0.5 heterojunction at source and pocket to improve band to band tunneling (BTBT) rate, at the same time, the gate electrode is divided into two parts namely tunnel gate (M1) and control gate (M2) with workfunctions ΦM1 and ΦM2, where ΦM1 > ΦM2. Thus, the proposed structure may have a great potential to boost the device performance of traditional vertical TFETs.

The reviewer has few notes which should be commented or amended by the authors to better reveal the manuscript results.

1) Figure 1 of Section 2 shows the transistor structures studied in the paper and Table 1 demonstrates the fundamental geometric parameters of these devices. However, the paper does not present sufficient ground to choose the values of these parameters. In reviewer’s opinion, it is necessary to reason the parameter values, presented in Table 1, in more details. The reviewer suggests to provide references (in Section 2) in which similar transistors are described and studied. Besides, it is necessary to discuss (in Section 2) key technological processes which the authors intent to use in order to fabricate the transistors with structure presented in Figure 1. The reviewer also suggests to describe restrictions and specifications of the technology for the transistor manufacturing.

2) In the paper, SiO2 is referred as a low-k dielectric (e.g. line #103). However, in microelectronics it is commonly accepted that dielectrics with dielectric permittivity lower than SiO2 case are treated as low-k dielectrics.

3) In Figures 2, 3, etc. in insets demonstrating the transistor structure the word “cutline” is typed with too low font size to be readable.

4) From the paper it is not clear which technological processes the authors intent to use to form gate dielectrics on the basis of HfO2 and SiO2 (technological processes, annealing, etc.). The manufacturing technology of the films can significantly influence on transistor characteristics and, thus, it is necessary to discuss it in the paper in more details.

Round 2

Reviewer 2 Report

Dears authors,

 First, I would like to appreciate the improvements introduced in this new version of your contribution after my suggestions. Now then, I still have some small comments to try to enhance your work.
 - Can you specify the deposition method of the HfO2 layer, please?
 - Comparing Fig 9 and 10, why do you present different graphs labeled as (c)? Which different information brings to your investigation each one?
 - A style comment, usually the figures should be mentioned as Fig. 10.c and not Fig. 10 (c).
 - From your comparison on using two different metal gate materials, which are your main conclusions? I think you should highlight it, to give more relevance to your work!
 - In my previous comments I mentioned you to highlight the increasing relevance of vertical topology. Therefore, I suggested you to check and take advantage of references 21 and 22. This references are related to other vertical device types and not to VTFET. I suggested you to talk about the current importance of vertical structures. For this I would like to suggest you to modify this small paragraph.

Reviewer 4 Report

Dear Authors,

Thank you for improving the paper quality.

I find only some minor issues concerning language or syntax. Please find my remarks below.

Several times phrases ", thus," ", i.e.," are used; I think there are too many commas; maybe it would be better to split the compound statements into simpler ones

on page 2

"In vertical TFET (VTFET), appropriate selection of work-function for metal electrodes can improve the trade-off between analog/RF and linearity parameters, thus, this device offers tremendous performance in terms of analog/RF and linearity results."; if a selection ofwork-function can improve, then the device can offer significant improvement of analog/RF performance and linearity; I doubt that you can call it tremendous (see Fig.12) 

"the VTFET is easier to manufacture than the conventional TFETs [16-20], where top-down nanofabrication technology can be used"; from this satement it results thatthe top-down technology is used in the case of conventional TFETs

on page 3

"Afterwards, deposit SiO2 by atom layer deposition" -> "Afterwards, SiO2 is formed by atom layer deposition"

on page 4

"Table 1. the fundamental" -> "Table 1. The fundamental"

on page 5

"In fact, tunneling is the carrier injection from source valance band to channel conduction band, the smaller valley of conduction band in source-pocket represents larger band to band tunneling without varying other conditions, i.e., the valley value of conduction band in source-pocket is minimal when y=0.5 in GaAsySb1-y, which makes a larger effective tunneling area compared with other conditions.";  this statement is awkward; tunneling is not the carrier injection from source valance band to channel conduction band; tunneling of electrons from source valance band to channel conduction band is the main transport mechanism in in TFETs; besides this statement is long; its 2nd part ", the smaller valley of conduction band..." is not clear

on page 6

"the current density of DMG-HD-VTFET is always larger than that of VTFET and HD-VTFET. As a result, the total current density of DMG-HD-VTFET is obviously enhanced, as shown in Figure 3. (d)"; from Fig.3d it results that the current density of DMG-HD-VTFET is larger in the given cross-section; by the way I guess that x-direction is perpendicular to the gate - it is not clear; I suggest also to write "... As a result, the total current density of DMG-HD-VTFET..."

on page 10

"when HP<6nm, In addition, the electric field" -> "when HP<6nm. In addition, the electric field"  
